# Intimacy and sexual functioning after cancer: The intersection with psychological flexibility

**Cecile J. Proctor[1], Anthony J. Reiman[2,3,4], Caroline Brunelle[1], Lisa A. Best[1] ***

**1** Department of Psychology, University of New Brunswick, Saint John, New Brunswick, Canada, **2** Department of Biological Sciences, University of New Brunswick, Saint John, New Brunswick, Canada, **3** Horizon Health Network, Saint John, New Brunswick, Canada, **4** Department of Medicine, Dalhousie University, Saint John, New Brunswick, Canada

* lbest@unb.ca

**Data Availability Statement:** The data is posted on Open Science Framework and data availability is noted in our paper. The doi is: DOI 10.17605/OSF. IO/FR5QT.

## Abstract

Cancer significantly impacts overall satisfaction with life (SWL). Psychological flexibility (PF) involves adapting to situational demands, balancing life demands, and committing to value-aligned behaviours, which can help survivors manage cancer-related distress. Given the lack of research examining how PF affects relationship and sexual satisfaction, our purpose was to elucidate the relationship between variables associated with partner intimacy, pillars of PF, and variables related to psychological wellness. We explored relationships between SWL and the pillars of PF (Valued Action, Behavioural Awareness, Openness to Experience) as mediating factors in the association between relationship and sexual satisfaction and SWL. Participants (113 male, 106 female) completed questionnaires measuring sexual function, intimacy, relationship and sexual satisfaction, PF, SWL, anxiety, and depression. Results indicated an equal percentage (57.5%) of males and females reported sexual dysfunction; however, varied patterns of relationships emerged between the sexes. There were significant relationships between SWL and relationship and sexual satisfaction. The mediation analyses showed that valued action and openness to experience partially mediated the relationship between relationship satisfaction and SWL. Interestingly, Valued Action was the only pillar of PF that emerged as a partial mediator between sexual satisfaction and SWL. Thus, value-aligned behaviours may be a key focus for intervention in cancer survivors.

## Introduction

Cancer can significantly impact the quality of life (QOL) of survivors and their intimate partner [1]. Survivors and their intimate partners behave in ways that can either strengthen or disrupt connections in marital relationships, and these interactions influence adjustment during treatment. Individual coping behaviours and mutual support can improve overall wellness during all cancer stages and can aid in adjustment and acceptance through periods of relapse [2]. As perceived closeness increases, the overall positive psychosocial adaptation to cancer improves [3]. Relationship factors, including sexual function and intimacy, significantly impact overall QOL; unfortunately, cancer, treatment, and side effects result in significant changes in sexual functioning, relationships, and even one's sense of self and identity [4].

**Funding:** This research was funding by the Social Sciences and Humanities Research Council Doctoral Scholarship (CP), the Canadian Cancer Society (AR), The Canadian Institute for Health Research/Research NB (AR), and Research NB (LB). The funders had no role in study design, data collection and analysis, decision to publish, or preparation of the manuscript.

**Competing interests:** The authors have declared that no competing interests exist.

Female survivors are six times more likely to report separation or divorce than male survivors [5]. Most couples adapt and persevere after cancer, but the cancer experience can strain relationships and lead to marital distress.

## Sexual dysfunction and intimacy after cancer

Feelings of isolation, anxiety, depression, or inadequacy may emerge due to changes in sexual functioning [2]. Sexual function is impacted by multiple factors, including side effects of surgery, medication, or treatment [6] and psychological factors, such as body image and emotional distress or low mood [7]. Hawkins found that 30% of men and 33% of women experienced physical barriers to traditional sexual intercourse [1]; 75% of female breast cancer survivors met the criteria for sexual dysfunction, and 71% of these women felt unprepared for the treatment's impact on their sexual function [8]. Given that illnesses, pain, anxiety, rage, stressful situations, and medication affect sexual functioning [9, 10], sexual function assessment is vital for survivors to identify treatment side effects and improve QOL [11]. There is a growing recognition that providers are not addressing these factors in individual treatment plans [10].

Intimacy is a multifaceted construct that increases emotional, social, sexual, intellectual, and recreational intimacy needs [12]. Intimacy is a primary psychological need that extends beyond sexual functioning [1]. Intimacy is not a destination but a process of growth that can change over time. Although more than half of survivors (62%) reported a desire for sexual intimacy, physical limitations impaired their sexual function [13]. Survivors need to realize that it is possible to develop and maintain an intimate relationship without sexual function [12]. According to Flynn et al. [14], most survivors experienced diminished intimacy post-cancer; less than half were satisfied with the sexual aspects of their relationships. Although essential, emotional intimacy is not sufficient for sexual satisfaction but can be interpreted as a reward of sexual activity. Sexual satisfaction is highly related to sexual function but is distinct from overall relationship satisfaction [14, 15]. The length of relationships impacts satisfaction and varies for males and females [16], supporting the need for research exploring the patterns of associations between sexual and relationship satisfaction in males and females.

Given the varied impact of sex-specific cancers on sexual functioning, examining sex differences in this population is vital. Reported disruptions include sex drive decreases, fear of initiating sex, difficulty recapturing "normality," and feeling unwanted or unattractive because of a lack of sex [2]. Cancer impacts women's sexual functioning and intimate relationships, resulting in feelings of inadequacy [17, 18], resentment, withdrawal from their partner, and overall relationship discord [19]. Feelings of obligation can push women into sexual activity despite their dissatisfaction [20], which can lead male partners of survivors to feel that their spouse is engaging in sexual activity despite a lack of desire/interest [20]. Many sexual function issues resulting from cancer persist over time [21].

For men, impacts go beyond erectile dysfunction and permeate into psychosocial domains; therefore, using partners in interventions relating to sexual problems is critical [22]. Long-term testicular cancer survivors reported changes in their intimate relationships, ranging from strengthened relationships for married men to strained relationships for single men with sexual partners [23]. Men reported feeling a sense of "failure" when unable to engage in sexual activities that they were previously able to perform. Male response and perception of sexual dysfunction or loss of sexual function vary. For couples in the survivorship phase, resuming a sexual relationship is challenging; however, many long-term testicular cancer survivors and their partners believe their cancer experience brings them closer [23].

## Psychological flexibility

Psychological Flexibility (PF) is amenable to change. It is the measured outcome of Acceptance and Commitment Therapy (ACT), a third-wave behavioural therapy that targets internal and external behaviours. There are three pillars of PF: (1) being consciously present in the moment (behavioural awareness); (2) being open to positive and negative experiences (openness to experiences); and (3) valued action (engaging in or changing behaviours to align behaviours with personal values). Changes in PF can occur by increasing resilience and teaching individuals to prevent negative thoughts and feelings from driving their behaviour [24]. Individuals with low PF have difficulty engaging with thoughts, emotions, and behaviours that align with personal values and may react judgmentally to internal and external experiences. Because the ACT process does not involve teaching individuals to dismiss negative thoughts, it can help individuals actively engage with their trauma and bolster self-awareness, preventing a cancer diagnosis from fully defining their sense of self [25]. Daily incorporation of psychological flexibility allows individuals to focus on the present moment and more effectively cope with trauma while engaging in meaningful lives [26]. It is important to understand PF as one factor of resilience that includes components beyond resiliency [27].

The ACT process encourages acceptance of positive and negative experiences but is not passive. Acceptance is an active reaction that intentionally shifts the focus from avoiding unwanted thoughts and feelings to moving toward them with interest, curiosity, and, most importantly, non-judgemental observation. Much like distress intolerance and emotional suppression, experiential avoidance offers short-term relief from discomfort with the cost of long-term adverse effects [28]. By incorporating acceptance, individuals may move beyond avoidance and accept negative experiences as a normal part of survivorship.

Moderate inverse correlations exist between psychological flexibility and depression, anxiety, anxiety sensitivity, behavioural inhibition, and the personality factors of neuroticism and extraversion (positive correlation;[29]). Depression, psychosis, and burnout have all been treated successfully with ACT through the improvement of PF [30]. Psychological flexibility significantly predicts functioning and impairment in clinical samples (Panic Disorder with Agoraphobia, Anxiety/Social Phobia), even when controlling for depressive symptoms, neuroticism, and anxiety sensitivity [31]. Generally, PF predicts levels of functioning and not specific symptomatology or diagnostic presentation.

After cancer, male and female survivors can experience a loss of sexual function; however, some people can renegotiate their intimacy to include sexual practices that were previously unexplored [1]. A psychologically flexible individual chooses values-based behaviours even when internal (for example, cognitions) or external (for example, pain, discomfort) obstacles are present [32]. Higher levels of PF are not expected to alleviate physical symptoms but are related to increased well-being and reduced distress [33]. Furthermore, increases in PF through ACT have been linked to improved marital adjustment and satisfaction [32]. Through ACT, PF can be improved and changed over time [24, 34].

When individuals fear or avoid intimacy, their ability to form and maintain relationships is hindered, and behaviours that conflict with emotional connection and intimacy may develop [32]. By avoiding intimacy, individuals are not fully accepting of their inner experiences, and they may avoid unpleasant inner experiences and even avoid behaviours that align with values associated with their relationship. This psychologically inflexible approach can lead to difficulty in solving relational disputes, holding grudges, or even a lack of forgiveness, which can strain relationships [32]. Exploring specific factors related to higher intimacy despite sexual dysfunction is an essential avenue of study.

## Purpose of the current study

Our objectives were to answer four specific research questions: 1. What levels of sexual dysfunction do male and female cancer survivors report? 2. Are relationship and sexual factors (sexual dysfunction, intimacy, sexual satisfaction, relationship satisfaction) related to the pillars of psychological flexibility? 3. Are relationship and sexual factors (sexual dysfunction, intimacy, sexual satisfaction, relationship satisfaction) related to mental health variables, including symptoms of anxiety and depression, and overall satisfaction with life (SWL) for male and female cancer survivors? 4. For male and female cancer survivors, how are relationship and sexual satisfaction related to SWL and the three pillars of PF?

## Method

### Participants

In this study, 219 cancer survivors were recruited through Prolific (https://www.prolific.co/), an online participant recruitment service. The sample included 106 females ($M_{age}$ = 53.39 years, $SD$ = 11.91) and 113 males ($M_{age}$ = 54.94 years, $SD$ = 15.40). All individuals reported a previous cancer diagnosis, were over 19 years old, and were currently in what *they* defined as a romantic relationship. For this study, we asked people to indicate their relationship status. Only individuals who reported being in a romantic relationship (in a relationship, married, in a civil partnership/civil union) were eligible to participate. We did not collect information on the gender of their partner.

### Materials

#### Acceptance and commitment therapy processes

The Comprehensive Assessment of Acceptance and Commitment Therapy Processes (CompACT; [34]) uses 23 items to measure an individual's PF, with higher scores indicating higher PF. Using a 7-point scale from 1 (*strongly disagree*) to 7 (*strongly agree*), individuals are presented three subscales that contain both negative and positively valenced statements: Openness to Experience (OE; for example, "Thoughts are just thoughts—they don't control what I do"); Behavioural Awareness (BA; for example, "I rush through meaningful activities without being attentive to them"); and Valued Action (VA; for example, "I can identify the things that matter to me in life and pursue them"). Internal consistency was satisfactory in the current study, ranging from α = .83 to .86.

#### The patient health questionnaire

The Patient Health Questionnaire (PHQ-9; [35]) uses nine items to classify depression symptoms that align with the nine DSM-IV criteria for major depressive disorder [35]. Questions such as "little interest or pleasure in doing things" are rated on a 4-point scale from nearly every day to not at all. This measure is valid and reliable and aids in the diagnosis of depression, and provides cut-off scores for symptom severity, with higher scores indicating more severe symptoms [35]. The current Cronbach's alpha was .90.

#### Generalized anxiety disorder 7-item

The Generalized Anxiety Disorder 7-item (GAD-7; [36]) measures seven symptoms related to generalized anxiety disorder and its severity (for example, "not being able to stop or control worrying"). The scale uses a 4-point Likert scale ranging from not at all nearly every day, with

scores greater than 10 indicating a clinically significant condition with a sensitivity of 89% and specificity of 82% [36]. In this study, the reliability was high, Cronbach's Alpha = .93.

## Changes in sexual functioning questionnaire

Sexual dysfunction was measured using the 14-item sex-matched versions (male/female) of the Changes in Sexual Functioning Questionnaire (SFQ; [37]). Participants completed the sex-specific questionnaire based on their reported sex assigned at birth. Some questions presented to males and females are different based on specific sexual function concerns related to each (for example, females: "Do you have adequate vaginal lubrication?"; males: "Are you able to maintain an erection?"). General questions unrelated to sex-specific function are included for males and females (for example, "How much pleasure or enjoyment do you get from your orgasms?"). We used the established cut-off threshold total scores for males ($<41$) and females ($<48$; [37]) to assess dysfunction. Cronbach's alphas for the individual subscales (sexual desire/frequency, sexual desire/interest, arousal/excitement, orgasm/completion) were acceptable, ranging from $\alpha = .72$ to $\alpha = .87$.

## Personal assessment of intimacy in relationships

The Personal Assessment of Intimacy in Relationships (PAIR; [12]) is a self-report measure that includes 36 items with five subscales (Emotional, Social, Intellectual, Sexual, and Recreational) capturing various domains of intimacy [12]. Questions are answered in relation to a romantic partner. This scale has been validated in studies of heterosexual [38, 39] and same-sex couples [39]. In the current study, Cronbach's alphas were between .70 and .77 for the subscales.

## Satisfaction with life scale

Participants completed the Satisfaction with Life Scale (SWLS; [40]) to measure subjective well-being. This five-item measure uses a 7-point Likert scale (1 = strongly disagree). Higher scores indicate higher satisfaction with life. Cut-off scores indicate scores $>25$ represent high satisfaction across life domains, and scores between 20 and 24 show a general satisfaction with life. Scores under 20 indicate dissatisfaction with at least one area of their life. It is typical for individuals with chronic illness to have lower than average life satisfaction. The current reliability was also high, with Cronbach's alpha = .92.

## Relationship and sexual satisfaction

The Global Measure of Relationship Satisfaction (GMREL; [16]) and the Global Measure of Sexual Satisfaction (GMSEX; [15]) issue five 7-point bipolar scales (*good-bad; pleasant-unpleasant; positive-negative; satisfying-unsatisfying; valuable-worthless*). These scales provide global evaluations of an individual's relationship's positive and negative dimensions. Higher scores indicate lower satisfaction; for interpretability, which we reversed when reporting results. In the current study, the Cronbach's alphas were high; GMSEX = .97 and GMREL = .96.

## Procedure

The University of New Brunswick Research Ethics Board (REB File #2022–162) reviewed this project. Participants were recruited via Prolific, in which participants are matched for specific studies based on inclusion criteria. Data was collected between 14 July 2023 and 30 July 2023. In the current study, we invited male and female participants over 19 years old who reported

English as their first language, had a previous cancer diagnosis, and were currently in an intimate relationship. All participants who met these inclusion criteria and completed all the measures were included in the analyses. Participants completed the questionnaire package after reading preliminary information and providing informed consent. They completed the demographic and disease-specific questions and were presented with the remaining questionnaires in randomized order. The survey completion time was approximately 12 minutes. After the survey was completed, all participants were paid £3 for their participation.

## Data analysis

Before data analysis, data was screened for missing values, and univariate and multivariate outliers were removed. Data conditioning ensured that assumptions for each statistical model were met. Confidence intervals were used in addition to p values to adjust for multiple comparisons when assessing the statistical significance of correlation coefficients. Significant *p* values are only noted in the tables if the confidence interval for the correlation did not include zero. The regression analyses included assumptions of normality, linearity, and homogeneity, assessed before data analysis.

## Results

The reported number of years since first diagnosis was similar for males ($M_{years}$ = 11.44 years, $SD$ = 8.98) and females ($M_{years}$ = 11.32 $SD$ = 9.38). Participants reported living in the United States of America (42.0%), United Kingdom (50.7%), Australia (2.7%), and Canada (2.3%). Participants reported a variety of cancer types. The most frequent cancers were breast (24.7%), melanoma (11.4%), and prostate (8.2%). Over half the sample (Males: 58.5%; Females: 56.5%) reported their cancer at a stage between 0–2 and were expected to live more than five years at diagnosis (Males: 62.8%; Females: 64.1%). Almost all (98%) of the sample reported receiving some kind of treatment. Most participants (Males: 84.1%; Females: 79.2%) did not report a cancer relapse at the time of the survey.

### Research question 1: Sexual dysfunction and intimacy in survivors

Our first goal was to determine levels of sexual dysfunction and intimacy in male and female survivors. We hypothesized that comparable to existing research, over half of the respondents would report sexual dysfunction as measured by scores below the cut-off scores for males (48) and females (41) on the SFQ. In this sample, 57.5% of males and females had scores that fell below their cut-off scores, indicating sexual dysfunction. Thus, even though there were differences in the average SFQ scores of males and females, the overall levels of dysfunction were identical. Thus, our prediction that sexual dysfunction would be equally reported by males and females was supported.

Levels of intimacy in relationships reported by males and females are reported in Table 1. There were no sex differences in emotional, intellectual, sexual, social, and recreational intimacy. Further, there were no sex differences in sexual ($p$ = .505) or relationship satisfaction ($p$ = .611), indicating that males and females had comparable levels of sexual and relationship satisfaction.

We also examined associations between relationship (PAIR: intimacy; GMREL: relationship satisfaction), sexual satisfaction (GMSEX: sexual satisfaction) and factors associated with sexual dysfunction (SFQ). For females, most PAIR subscales were positively related to SFQ subscales, GMSEX, and GMREL scores. For males, all PAIR subscales were positively associated with GMSEX and GMREL. SFQ: pleasure was positively related to all PAIR subscales but other associations between PAIR and SFQ subscales were inconsistent. For example, PAIR:

Table 1. T-test comparisons between male (n = 113) and female (n = 106) participants.

| Variable | Males (*n* = 113) | Females (*n* = 106) | *t* (*p*) |
|---|---|---|---|
| | Mean (sd) | Mean (sd) | |
| **Sexual Function (SFQ-Total)**\* | 44.76 (10.64) | 39.42 (10.17) | -3.80 (. < .001) |
| **Intimacy (PAIR)** | | | |
| Emotional | 27.91(6.93) | 27.17 (8.15) | -.727 (.468) |
| Social | 23.00 (6.59) | 21.47 (7.91) | -1.551 (.122) |
| Intellectual | 28.30 (6.16) | 27.81 (6.97) | -.562 (.575) |
| Sexual | 22.23 (4.40) | 22.30 (4.83) | .104 (.917) |
| Recreational | 27.03 (5.96) | 26.27 (6.60) | -.888 (.375) |
| **Depression (PHQ-9)** | 14.45 (5.41) | 16.11 (5.93) | 2.167 (.031) |
| **Anxiety (GAD-7)** | 5.30 (5.41) | 6.85 (4.87) | 2.233 (.027) |
| **Satisfaction with Life (SWLS)** | 22.04 (6.70) | 20.85 (7.75) | -1.192 (.234) |
| **CompACT Subscales** | | | |
| Openness | 35.35 (9.86) | 32.28 (10.83) | -2.196 (.029) |
| Valued Action | 35.58 (6.42) | 35.96 (5.68) | .461 (.645) |
| Behavioural Awareness | 19.39 (6.40) | 17.88 (6.79) | -1.707 (.089) |
| CompACT Total Score | 90.33 (18.57) | 86.12 (18.99) | -1.660 (.098) |
| **Relationship Satisfaction (GMREL)** | 10.96 (6.05) | 11.44 (7.68) | .510 (.611) |
| **Sexual Satisfaction (GMSEX)** | 17.58 (8.46) | 16.74 (9.85) | -.668 (.505) |
| **Age** | 54.94 (15.40) | 53.39 (11.91) | -.830 (.407) |
| **Years Since Diagnosis** | 11.44 (8.98) | 11.32 (9.39) | -.098 (.922) |

*Note*. \* Separate but equivalent scales were used for male and female participants.

sexual was positively correlated with most SFQ subscales but SFQ: desire/interest was not associated with any PAIR subscales. The magnitude of these correlations was low to moderate (see Table 2).

We used *r-to-z* transformations to assess gender differences in the strength of the correlations between sexual function and intimacy variables. Except for the correlation between PAIR: Emotional and GMREL, $z = 2.68$, $p < .001$, which was significantly stronger for males than for females, the magnitude of the correlations was generally stronger for female participants than for male participants (see shaded cells in Table 2).

## Research question 2: Sex function, intimacy, and psychological flexibility

Our second research question was: Are relationship (PAIR: intimacy; GMREL: relationship satisfaction) and sexual factors (SFQ: sexual function; GMSEX: sexual satisfaction) related to the pillars of psychological flexibility (CompACT: OE, BA, VA)? For females, SFQ: Pleasure and SFQ: Orgasm was related to all three pillars of PF and CompACT: total; GMSEX was significantly correlated with Compact: BA, CompACT: VA, and CompACT: total. For males, the only significant relationship that emerged was between the SFQ: Pleasure and CompACT: VA. GMSEX was significantly correlated with CompACT: VA and Compact: total (see Table 3).

For females, all subscales of the PAIR, except PAIR: social and GMREL, were significantly correlated with the CompACT subscale and total score (see Table 3). For males, PAIR: emotional, intellectual, and recreational were significantly associated with the CompACT subscales and total score. PAIR: sexual was correlated with CompACT: VA. Interestingly, PAIR: social and GMREL were not associated with CompACT subscales or total scores for males. We used

**Table 2. Correlations for males and females between the sexual function and intimacy variables.**

| | SFQ: Pleasure | SFQ: Desire/ Frequency | SFQ: Desire/ Interest | SFQ: Arousal/ Excitement | SFQ: Orgasm | SFQ: Total | Sexual Satisfaction (GMREL) | Relationship Satisfaction (GMREL) |
|---|---|---|---|---|---|---|---|---|
| **Intimacy (PAIR)** | | | | | | | | |
| **FEMALES** | | | | | | | | |
| Emotional | .499*** | .274** | .159 | .221* | .484*** | .434*** | .662*** | .699*** |
| Social | .340*** | .204* | .147 | .253** | .289*** | .302*** | .405*** | .407*** |
| Intellectual | .525*** | .520*** | .223* | .254** | .443*** | .441*** | .650*** | .655*** |
| Sexual | .608*** | .335*** | .332*** | .325*** | .528*** | .581*** | .688*** | .410*** |
| Recreational | .495*** | .358*** | .242* | .322*** | .361*** | .422*** | .490*** | .578*** |
| **Relationship Satisfaction** (GMREL) | .460*** | .258** | .031 | .118 | .373*** | .304*** | .621*** | |
| **Sexual Satisfaction** (GMREL) | .759*** | .528*** | .296*** | -.395*** | .695*** | .676*** | | |
| **MALES** | | | | | | | | |
| Emotional | .365*** | .078 | -.043 | .112 | .105 | .146 | .282*** | .844*** |
| Social | .315*** | -.038 | -.007 | .069 | .030 | .074 | .204* | .432*** |
| Intellectual | .333*** | .102 | -.012 | .261** | .248** | .255** | .355* | .731*** |
| Sexual | .504*** | .223** | .094 | .302*** | .207* | .303*** | .466*** | .322*** |
| Recreational | .365*** | .147 | .090 | .204* | .126 | .210* | .261** | .623*** |
| **Relationship Satisfaction** (GMREL) | .370*** | .018 | .073 | .109 | .105 | .122 | .373*** | |
| **Sexual Satisfaction** (GMREL) | .771*** | .433*** | .211* | .603*** | .645*** | .671*** | | |

Note:

***p < .001,

** p < .01

*p < .05.

Shaded cells indicate significant differences in the strength of the relationships for males and females.

*r* to *z* transformations and determined no significant differences in the magnitude of correlations between the pillars of PF and intimacy, sexual dysfunction, sexual satisfaction, or relationship satisfaction between males and females. Because there were no differences in the magnitude of relationships for men and women, our mediation analyses included all participants to address research question 4.

## Research question 3: Sex function, intimacy, and psychological wellness

Research question 3 was, Are relationship (PAIR: intimacy; GMREL: relationship satisfaction) and sexual factors (SFQ: sexual function; GMSEX: sexual satisfaction) related to symptoms of anxiety (GAD-7) and depression (PHQ-9) and satisfaction with life (SWLS) for male and female cancer survivors? For males, SFQ total scores were not significantly correlated with GAD-7, PHQ-9, or SWLS. For females, SFQ total scores were significantly positively related to SWLS. Further, for females, GMSEX was significantly negatively associated with PHQ-9 and GAD-7, but positively with SWLS scores, but for males, GMSEX was only significantly positively related to SWLS (see Table 4).

For females, GAD-7 and PHQ-9 were significantly negatively correlated with PAIR: emotional, PAIR: intellectual, and PAIR: sexual. For males, PAIR: emotional and PAIR: intellectual were negatively related to GAD-7 and PHQ-9. Except for PAIR: social, all PAIR subscales were

**Table 3. Pearson product moment correlations between relationship and sexual variables and subscales of the CompACT.**

| | Openness to Experience (CompACT: OE) | | Behavioural Awareness CompACT: BA | | Valued Action CompACT: VA | | Psychological Flexibility CompACT: Total | |
|---|---|---|---|---|---|---|---|---|
| | Males | Females | Males | Females | Males | Females | Males | Females |
| **Intimacy (PAIR)** | | | | | | | | |
| Emotional | .228** | .236** | .288*** | .220* | .443*** | .299*** | .373*** | .304*** |
| Social | .088 | .300*** | .135 | .292** | .160 | .117 | .148 | .311*** |
| Intellectual | .309*** | .268** | .265*** | .266** | .364*** | .277** | .388*** | .332*** |
| Sexual | .101 | .267** | .076 | .212* | .252** | .238** | .167* | .300*** |
| Recreational | .203** | .232** | .244** | .196* | .396*** | .234** | .328*** | .273*** |
| **Sexual Functioning (SFQ)** | | | | | | | | |
| Pleasure | .027 | .226* | -.006 | .200* | .236** | .269** | .094 | .281** |
| Desire: Frequency | -.014 | .083 | -.036 | .015 | .147 | .048 | .031 | .067 |
| Desire: Interest | -.050 | .040 | -.121 | -.048 | .048 | .018 | -.052 | .011 |
| Arousal: Excitement | -.058 | .104 | -.094 | .058 | .050 | .027 | -.046 | .088 |
| Orgasm | .003 | .214* | -.037 | .231* | .135 | .306*** | .036 | .296** |
| **SFQ-Total** | -.019 | .190 | -.073 | .137 | .137 | .210* | .012 | .220* |
| **Sexual Satisfaction (GMSEX)** | -.011 | .234* | .017 | .245** | .204 | .278** | .071 | .304*** |
| **Relationship Satisfaction (GMREL)** | .213* | .145 | .216* | .265** | -.458*** | -.238** | .347*** | .249** |

Note:

***p < .001,

** p < .01

*p < .05.

positively related to the SWLS scores for both males and females. GMREL was significantly negatively associated with GAD-7, PHQ-9, and SWLS for both males and females.

We used *r* to *z* transformations and determined no significant differences in the magnitude of these correlations. There were significant differences in the magnitude of the relationships between PHQ-9 and SFQ: orgasm, *z* = 2.4, *p* = .016, and GMSEX, *z* = 2.9, .0037. There were also significant differences between GAD-7 and PAIR: sexual, *z* = 2.12, *p* = .034, and GMSEX, *z* = 2.2, *p* = .028. Finally, the magnitude of the correlation between SWLS and Orgasm stronger for females than males, *z* = 2.08, *p* = .038 (See Table 4).

## Research question 4: Sexual, relationship, and life satisfaction

To further investigate the different relationships between SWLS, GMREL, and GMSEX for the whole sample (*N* = 215), we conducted two mediation analyses using Hayes PROCESS [41] controlling for sex, age, relapse, and years since diagnosis. We entered the three pillars of psychological flexibility, CompACT: VA; CompACT: BA; CompACT: OE, as mediators in the relationship between GMREL and SWLS for Model 1 (see Fig 1) and between GMSEX and SWLS for Model 2 (see Fig 1). In Model 1, the indirect effects of Compact: VA [B = -.0778., CI: -.2105, -.0657] and Compact: OE [B = -.0500., CI: -.0576, -.0001] were statistically significant, accounting for 39.29% of the total relationship and partially mediated the relationship between GMREL and SWLS. In Model 2, the indirect effect of Compact: VA [B = -.0768., CI: -.1387, -.0412] was statistically significant accounting for 38.81% of the total relationship, partially mediating the relationship between GMREL and SWLS.

**Table 4. Correlations between relationship and sexual variables and mental health outcomes.**

| | Depression | | Anxiety | | Satisfaction with Life | |
|---|---|---|---|---|---|---|
| | (PHQ-9) | | (GAD-9) | | (SWLS) | |
| | Males | Females | Males | Females | Males | Females |
| **Intimacy (PAIR)** | | | | | | |
| Emotional | -.278** | -.307*** | -.291** | -.249** | .424*** | .454 *** |
| Social | -.047 | -.124 | -.113 | -.168 | .263** | .181* |
| Intellectual | -.266** | -.264** | .263** | -.215* | .324*** | .446*** |
| Sexual | .009 | -.243** | .043 | -.224* | .312*** | .329*** |
| Recreational | -.115 | -.158 | -.154 | -.110 | .357** | .423*** |
| **Sexual Functioning (SFQ)** | | | | | | |
| Pleasure | .014 | .258** | -.064 | -.183 | .372*** | .412*** |
| Desire: Frequency | .032 | -.005 | -.135 | .000 | .147 | .250 ** |
| Desire: Interest | .128 | .088 | .212* | -.030 | .048 | .110 |
| Arousal: Excitement | .088 | .028 | -.094 | .041 | .050 | .073 |
| Orgasm | .022 | -.299** | .098 | -.160 | .135 | .399*** |
| **Total** | .056 | -.164 | .138 | -.101 | .154 | .340*** |
| **Sexual Satisfaction (GMSEX)** | -.002 | .378*** | .048 | .250** | .301*** | .456*** |
| **Relationship Satisfaction (GMREL)** | -.257** | -.317*** | -.266** | -.264** | .445*** | .431*** |

Note:

***p < .001,

** p < .01,

*p < .05

Shaded cells indicate significant differences in the strength of the relationships for males and females.

## Discussion

Sexuality and intimacy are important components of QOL for cancer survivors and can be significantly affected by both disease and treatment [13, 14, 21]. Research shows that survivors often experience adverse changes in their sexuality related to the impact of cancer diagnosis and treatment [14]. These changes in sexual functioning affect not only QOL but also the quality of intimate relationships. This impact can range from changes in sexual desire and satisfaction to physical limitations and body image concerns. Furthermore, the incidence of sexual dysfunction among female cancer survivors has been reported to range from 30% to 90% across multiple datasets [42].

In the current study, levels of sexual dysfunction were high among both male and female survivors, with 57.5% of respondents reporting scores above the cut-off for sexual dysfunction. As hypothesized, there were no significant differences in the number of males and females reporting sexual dysfunction. This finding is consistent with previous research showing a high prevalence of sexual dysfunction in both genders after cancer [43]. These rates of sexual dysfunction are alarming and highlight the significant impact that it can have on individuals' lives. We also examined levels of intimacy in male and female survivors and found that although females reported lower levels of social intimacy than males, there were no gender differences in emotional, intellectual, sexual, and recreational intimacy.

Sexual problems are difficult to predict after a cancer diagnosis. They can occur in those without other psychosocial risk factors and in any area of sexuality at any time, from pre-treatment to post-treatment [6, 21, 35]. Because there is not enough information about the effects of cancer on sexual relationships, it is challenging to advise therapists on how to help clients

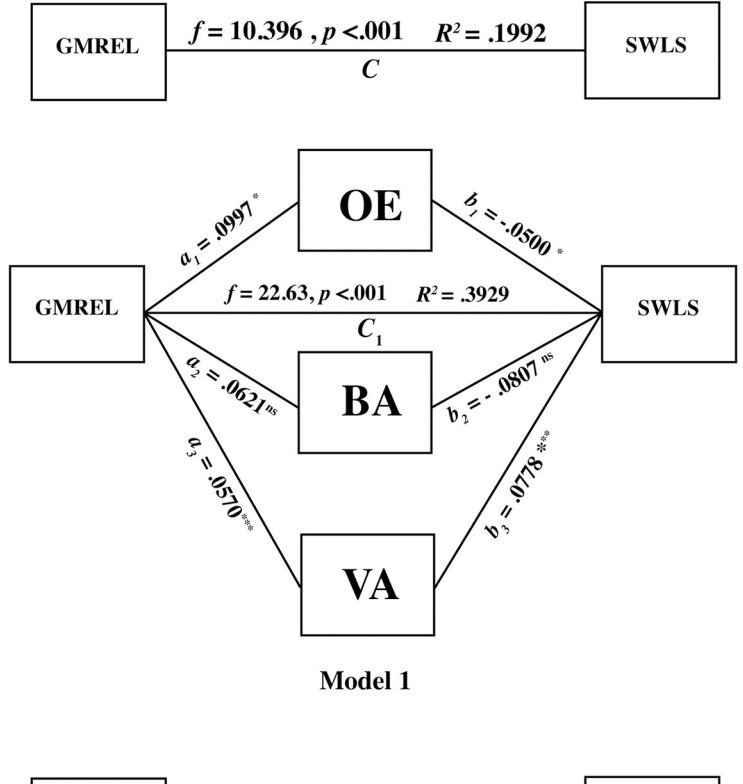

**Model 1**

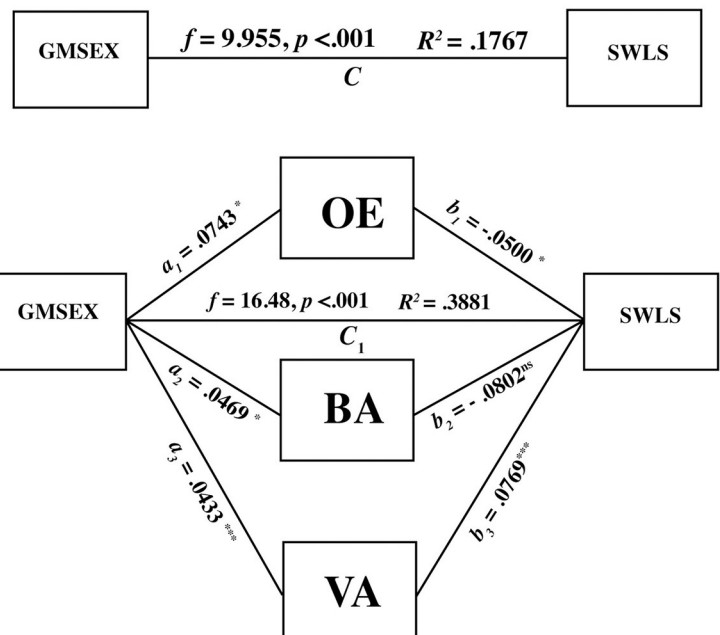

**Model 2**

**Fig 1. Model 1 and Model 2.**

best. A better understanding of which survivors are most at risk of having a negative impact on sexuality could benefit therapists and other health professionals [43].

## Psychological flexibility and its connection to relationship and sexual factors

Our second research question focused on the associations between relationship and sexual factors (i.e., dysfunction, intimacy, sexual and relationship satisfaction) and the pillars of PF. The results of the current study indicated noteworthy gender differences in the associations between sexual and relationship factors and PF. For females, relationship satisfaction was significantly correlated with all three pillars of PF. On the other hand, for males, the only significant relationship that emerged was between sexual pleasure and PF: valued action. Also, sexual satisfaction was significantly correlated with PF: valued action and overall psychological flexibility for males in this sample. Considering these results, female patients may benefit from all aspects of ACT while male clients may benefit from introducing value identification and goals related to those values early on in treatment.

For females, sexual pleasure and orgasm were related to all three pillars of PF, supporting Bahrami et al. [44], who reported that women's sexual arousal and satisfaction are closely tied to the quality of their relationship. This is important to note for clinicians working with female cancer patients; fostering PF may help to increase sexual pleasure and, in turn, improve relationship quality and satisfaction with life. The discrepancies between male and female reports highlight the importance of considering sex differences when examining the relationship between sexual and relationship factors and PF.

## Relationship and sexual factors and psychological well-being

We also examined whether relationship and sexual factors, such as sexual dysfunction, intimacy, sexual and relationship satisfaction, were related to symptoms of anxiety and depression and satisfaction with life for male and female survivors. Current results indicated that for female survivors, there was a significant relationship between sexual function (measured by the SFQ total scores) and life satisfaction (as measured by the SWLS); however, for male survivors, correlations between sexual factors and symptoms of anxiety and depression and overall life satisfaction were not statistically significant. These relationships do not infer causality; therefore, it is possible that, for females, when life presents challenges leading to lower life satisfaction and increased anxiety and depression, sexual function is impacted but it is also possible that females with higher levels of psychological distress may experience sexual dysfunction. From a clinical perspective, for females, these lower levels of sexual functioning may be an indication that their mental health is suffering. Thus, it is possible that targeting one may improve the other.

For males, only relationship satisfaction and intimacy were related to overall satisfaction with life. In Flynn et al.'s [14] qualitative study, some male respondents described a tendency to push away from their partners with an increased desire to be alone, citing shame and failure as feelings associated with the changes in sexual function. It may be that this "pushing away" mentioned by Flynn and their colleagues contributes to overall distance in relationships. Previous research has shown that many long-term testicular cancer survivors and their wives believe their cancer experience brings them closer [23], indicating that emotional factors beyond sex may enhance relationship factors even when sexual satisfaction is low.

Thus, for males, there appears to be a distinction between relationship/emotional factors and sexual satisfaction, whereas, for females, both components were linked to satisfaction with life. Further investigation may reveal whether it is the sexual components of relationships that

impact SWL and mental health (i.e., anxiety, depression) for females or if these other factors, such as mental health, intimacy, and SWL, affect their sexual desire, engagement, and pleasure. Further studies should use qualitative approaches to examine these complex relationships.

In line with previous research, our results revealed that for female survivors, symptoms of anxiety and depression (including GAD-7 and PHQ-9 scores) were significantly correlated with emotional, intellectual, and sexual aspects of intimacy (as measured by the PAIR sub-scales), suggesting that sexual and relationship factors play a significant role in their overall well-being and satisfaction with life. Cancer significantly impacts women's sexuality, sexual functioning, intimate relationships, and sense of self [17, 18]. These feelings can lead to resentment, withdrawal from their partner, and discord [19]. Overall, factors such as sexual dysfunction, lower intimacy, and sexual and relationship dissatisfaction are associated with symptoms of anxiety and depression, as well as lower overall satisfaction with life, for female cancer survivors. The current results highlight the importance of addressing sexual factors to enhance female cancer survivors' overall satisfaction and QOL. Further, if practitioners see sexual difficulties in female survivors, they should address the possibility that these difficulties may be linked to mental health difficulties or rooted in relationship distress. This aligns with previous research suggesting that relationship distress significantly relates to sexual functioning in breast cancer survivors [44].

## Sexual satisfaction, satisfaction with life and psychological flexibility

Our final mediation analyses showed that the indirect effects of *valued action* and *openness to experience* were statistically significant, partially mediating the relationship between *relationship satisfaction* and SWL. Thus, engaging in values-based behaviours and being open to positive and negative experiences helped explain the link between *relationship satisfaction* and overall SWL. This aligns with previous research by Swash et al. [24], who reported that increased psychological flexibility is associated with lower levels of psychological distress and higher QOL. For *sexual satisfaction*, the indirect effect of *valued action* was the only statistically significant mediator, partially mediating the relationship between sexual satisfaction and satisfaction with life. This value-based behaviour helps explain the link between *sexual satisfaction* and SWL. Regarding sexual satisfaction and SWL, a dedication to engage in sexual behaviours because that connection is something valued in a relationship may be a vital component. These findings provide evidence for the role of psychological flexibility, specifically *valued action* and *openness to experiences*, in the relationship between general mental health related to relationship quality and subjective well-being with life satisfaction.

Valued action emerged as the strongest mediator of both sexual and relationship satisfaction. When encouraging value-aligned behaviours, values are defined as not something to obtain, but instead, values are personal and focus on the direction of one's life and personal motivations to move forward [45]. For example, a survivor may hold "connection" as a core value in their relationship and, thus, must work towards participating in behaviours and actions that align with those values (for example, engaging in intimate acts despite physical obstacles to traditional sexual activity). The openness component of the ACT process, which also emerged as a mediator, is not focused on eliminating or changing negative thoughts; instead, an individual may have persistent negative thoughts and yet be able to interact flexibly with both positive and negative inner experiences to live a life that is in line with their core values [29]. Behavioural awareness involves being present in the moment and enhances one's ability to engage in moment-to-moment behaviours that align with personal values [46]. This did not emerge as a mediator, indicating that the acceptance of positive and negative experiences and value-aligned behaviour are the most effective combination in relationships. The

most important part of ACT is getting to those behaviours. For this sample openness to positive and negative experiences and the behaviours and values themselves contributed to overall satisfaction with life.

## Strengths and limitations of the current study

The participants reported a variety of cancer types, a wide range of time since diagnosis, and included those who had experienced relapse and those who did not. This created a well-represented pool of participants to answer our research questions but did not allow us to examine differences associated with different types of cancer. Because our research questions were novel, the relationships we examined were broad. A limitation to this approach is that we did not collect in-depth information on specific aspects of intimate relationships, such as length of relationships, living arrangements, or gender of partner. The benefit of this approach is that we were able to narrow our research questions to target the process-level relationships but, at the same time, we may have missed important factors, such as and length of relationship, ages of partners, and relationship history.

Online recruitment has become increasingly popular in health research due to its ability to reach a broad and diverse population rapidly and cost-effectively [47, 48]. This approach is advantageous and can help researchers address common enrollment challenges, such as reaching individuals, including cancer survivors, who are typically difficult to recruit [49]. At the same time, online recruitment does limit the range of participants who can participate as data collection is limited by internet access. The majority of participants were from the United States and the United Kingdom, which may limit the generalizability to other geographic locations.

## Implications

Healthcare professionals may overlook discussions about sexual health and functioning during cancer treatment and in the years post-treatment. This lack of communication can leave survivors feeling unsupported and uninformed about the potential changes they may experience in their sexual and intimate relationships. As a result, healthcare providers need to initiate conversations about sexual dysfunction relationship satisfaction and provide appropriate support and resources for cancer survivors. Focusing on values-based approaches and accepting positive and negative experiences will enhance the likelihood of relationships and life satisfaction.

## Conclusion

Sexual health is an essential for individuals after a cancer diagnosis, and the treatments for these cancers can adversely affect sexual health. In conclusion, the findings of this study highlight the mediating role of personal functioning in the relationship between relationship satisfaction and life satisfaction. Specifically, the pillars of valued action and openness to experience played a significant role in this mediation process.

## Supporting information

**S1 File.**
(DOCX)

## Author Contributions

**Conceptualization:** Cecile J. Proctor, Anthony J. Reiman, Caroline Brunelle, Lisa A. Best.

**Formal analysis:** Cecile J. Proctor, Lisa A. Best.

**Funding acquisition:** Cecile J. Proctor, Lisa A. Best.

**Investigation:** Cecile J. Proctor.

**Methodology:** Cecile J. Proctor, Anthony J. Reiman, Caroline Brunelle, Lisa A. Best.

**Project administration:** Cecile J. Proctor.

**Resources:** Lisa A. Best.

**Supervision:** Lisa A. Best.

**Visualization:** Cecile J. Proctor.

**Writing – original draft:** Cecile J. Proctor.

**Writing – review & editing:** Cecile J. Proctor, Anthony J. Reiman, Caroline Brunelle, Lisa A. Best.

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
