## [Editor Report · Decision Letter 0]

29 Nov 2023

PMEN-D-23-00021

Intimacy and sexual functioning after cancer: The intersection

with psychological flexibility

PLOS Mental Health

Dear Dr. Best,

Thank you for submitting your manuscript to PLOS Mental Health. After careful consideration, we feel that it has merit but does not fully meet PLOS Mental Health’s publication criteria as it currently stands. Therefore, we invite you to submit a revised version of the manuscript that addresses the points raised during the review process.

We look forward to receiving your revised manuscript.

Kind regards,

Akash Kumar Mahato, Ph.D.

Academic Editor

PLOS Mental Health

Journal Requirements:

1.Please provide separate figure files in .tif or .eps format.

https://journals.plos.org/mentalhealth/s/figures 

https://journals.plos.org/mentalhealth/s/figures#loc-file-requirements 

Additional Editor Comments (if provided):

1. Page 12 and 13, the section on implication of Acceptance and commitment therapy, gives an impression that the participants underwent some therapeutic program and does not match as per research objectives stated in the abstract and is unwarranted.

2. Conceptualizing description of the psychological variables (Psychological flexibility) may be expanded. Whether it is same or different with the concept of resilience, and how the concept has been assessed in available literature.

3. In methods section please write full name of the tools used in header (ACT process).

4. Exclusion criteria if any in brief need to be mentioned.

5. Page 15- name of rating scale should be Generalized Anxiety Disorder- 7 item

6. Reference 8. check format
---

## [Decision Letter · Decision Letter 1]

22 Feb 2024

PMEN-D-23-00021R1

Intimacy and sexual functioning after cancer: The intersection

with psychological flexibility

PLOS Mental Health

Dear Dr. Best,

Thank you for submitting your manuscript to PLOS Mental Health. After careful consideration, we feel that it has merit but does not fully meet PLOS Mental Health’s publication criteria as it currently stands. Therefore, we invite you to submit a revised version of the manuscript that addresses the points raised during the review process.

Please submit your revised manuscript by . If you will need more time than this to complete your revisions, please reply to this message or contact the journal office at mentalhealth@plos.org. Please include the following items when submitting your revised manuscript:

We look forward to receiving your revised manuscript.

Kind regards,

Akash Kumar Mahato, Ph.D.

Academic Editor

PLOS Mental Health

Journal Requirements:

1. In the online submission form, you indicated that [Data is available, by request, from the corresponding author.]. 

3. Uploaded as supplementary information.

Additional Editor Comments (if provided):

Reviewers' comments:

Reviewer's Responses to Questions

**Comments to the Author**

1. If the authors have adequately addressed your comments raised in a previous round of review and you feel that this manuscript is now acceptable for publication, you may indicate that here to bypass the “Comments to the Author” section, enter your conflict of interest statement in the “Confidential to Editor” section, and submit your "Accept" recommendation.

Reviewer #1: All comments have been addressed

2. Does this manuscript meet PLOS Mental Health’s publication criteria? Is the manuscript technically sound, and do the data support the conclusions? The manuscript must describe methodologically and ethically rigorous research with conclusions that are appropriately drawn based on the data presented.

Reviewer #1: Partly

3. Has the statistical analysis been performed appropriately and rigorously?

Reviewer #1: Yes

4. Have the authors made all data underlying the findings in their manuscript fully available (please refer to the Data Availability Statement at the start of the manuscript PDF file)?

Reviewer #1: No

5. Is the manuscript presented in an intelligible fashion and written in standard English?

Reviewer #1: Yes

6. Review Comments to the Author

Reviewer #1: See the attachment for comments to the authors

7. PLOS authors have the option to publish the peer review history of their article (what does this mean?). If published, this will include your full peer review and any attached files.

**Do you want your identity to be public for this peer review?** For information about this choice, including consent withdrawal, please see our Privacy Policy.

Reviewer #1: No

---

## [Editor Report · Decision Letter 2]

30 Apr 2024

Intimacy and sexual functioning after cancer: The intersection with psychological flexibility

PMEN-D-23-00021R2

Dear Dr Best,

We are pleased to inform you that your manuscript 'Intimacy and sexual functioning after cancer: The intersection with psychological flexibility' has been provisionally accepted for publication in PLOS Mental Health.

Best regards,

Akash Kumar Mahato, Ph.D.

Academic Editor

PLOS Mental Health